# Experiences with Participation in a Supervised Group-Based Outdoor Cycling Programme for People with Mental Illness: A Focus Group Study

**DOI:** 10.3390/ijerph16040528

**Published:** 2019-02-13

**Authors:** Helle Schnor, Stina Linderoth, Julie Midtgaard

**Affiliations:** 1University College Copenhagen, 2200 Copenhagen N, Denmark; 2Mental Health Centre Glostrup, 2600 Glostrup, Denmark; stina.rose.linderoth@regionh.dk; 3The University Hospitals Centre for Health Research, Copenhagen University Hospital Rigshospitalet, 2100 Copenhagen Ø, Denmark; julie@ucsf.dk; 4Department of Public Health, University of Copenhagen, 2200 Copenhagen N, Denmark

**Keywords:** mental illness, outdoor cycling, community mental healthcare, physical activity, health promotion, focus group

## Abstract

Epidemiological evidence suggests that physical exercise, notably popular sports, is associated with reduced, mental health burden. This study explored participation in a supervised, group-based, outdoor cycling programme (10 × 10 km rides over a five-month period) for people with mental illness. We conducted two rounds of three audio-taped focus groups with people with mental illness (*n* = 25, mean age = 40 years) that focused on previous physical activity and motivation for enrolment (baseline), and on programme evaluation, including subjective wellbeing (after 10 weeks). Transcribed verbatim, the group discussions were analysed using systematic text condensation, which identified 12 categories and four themes: 1) Reinvigoration, (2) motivation through equal status, (3) group commitment without focus on illness, and (4) the value of cycling. Of particular interest was the potential for outdoor cycling to support unique non-stigmatising therapeutic relationships in a non-patient environment, outdoor sensory experiences, e.g., fresh air, wind, and rain, and feelings of personal mastery, equal status, solidarity, community, and healing. This study indicated that outdoor cycling performed in groups supervised by healthcare staff may support exercise self-efficacy and empower people with mental illness, potentially promoting long-term physical activity and participation. Future interventional studies examining the effectiveness of outdoor cycling complementary to conventional community mental healthcare services are warranted.

## 1. Introduction

It has been well documented that people with severe mental illness have a shorter life expectancy compared to the general population [1,2,3,4]. The risk of increased mortality is mostly associated with physical health conditions, such as cardiovascular disease, respiratory disease, and cancer. These conditions are significantly related to increased body weight, hypertension, dyslipidemia, and insulin resistance in people with severe mental illness, in addition to suboptimal integration of somatic and psychiatric health care services [5,6]. The side effects of antipsychotic medication associated with weight gain and metabolic syndrome correspondingly contribute to cardiovascular risk and excess mortality [7,8,9].

One review shows that the impact of physical exercise on people with serious mental illness varies depending on the intervention and what parameters are measured. In the included quantitative studies, there were no obvious changes in mental illness symptoms, body mass index, or body weight [10]. However, a recent large (*n* = 1.2 million individuals) cross-sectional study concluded that physical exercise was significantly and meaningfully associated with reduced self-reported mental health burden [11]. Specifically, the authors showed that engagement in popular sports, mostly team based, was associated with the lowest mental health burden, and argued that this may be related to the promotion of resilience to stress and reduction in depression, due to participation in a social activity [11].

However, participation in physical activity for people with mental illness is lower than in the general population. Low self-efficacy is one of the strongest determinants for non-participation and there is a gap between interest and the capacity to do physical activity [12,13]. Additional known impediments to participation in physical activity include social isolation, fatigue, lack of energy, and overweight [9,14,15,16]. Moreover, people with mental illness as well as somatic comorbidity, typically cardiovascular diseases, find it even more difficult to engage in physical activity due to the experience of pain [15,16]. By contrast, those people who already feel in good shape are less inhibited about participation [9,14,15,16]. A qualitative literature review summarised the perspective of users of mental health services and found that they experience physical activity as socially inclusive, non-stigmatising, and effective in aiding recovery [17]. Moreover, a recent systematic review and meta-analysis showed reduced symptoms in people with mental illness and improved quality of life in relation to participation in physical activity interventions [18].

In spite of scientific studies suggesting that physical activity is a highly acceptable and efficacious component of care, it is not yet implemented as a routine part of psychiatric care [19].

Outdoor cycling is a popular form of physical exercise that can be done at different levels of intensity and is also the most widespread form of physical activity in Denmark, where bicycles are used on a daily basis as a means of transport and physical exercise. Furthermore, cycling is a flexible pursuit that predominantly takes place outdoors and, in some cases, in the company of others. New studies show that a brief bike ride every day can bring significant health benefits [20,21], including a reduction in the risk of cancer, cardiovascular mortality, and overweight among middle-aged adults [22].

To our knowledge, only a single study has previously examined the impact of outdoor cycling as a means to health promotion in people with mental illness. The study explored the feasibility of group-based cycling over a three-month period in patients with schizophrenia. It demonstrated an improvement in cardiovascular fitness and maintenance of this level after six months, as well as lasting improvements in social relations [23]. Moreover, engaging in cycling has been shown to be associated with low mental health burden, both in the general population and in people with a previous diagnosis of depression [11]

Against this background, and with the aim of increasing general awareness of the possibility of improving the health of people with mental illness via physical exercise, the Danish Workers’ Sports Confederation and Danish Cyclists’ Federation, together with the Danish Mental Health Fund and Danish Association for Mental Health organised for the second time the nationwide programme called “100K for Mental Health” in 2017. The programme was held in collaboration with community mental healthcare centres in eight towns and cities in Denmark and offered supervised 10 km bike rides, weekly.

The aim of the current study was to explore the experiences of people with mental illness who took part in group-based outdoor cycling, including their experience of physical, social, and mental wellbeing.

## 2. Materials and Methods

### 2.1. Design

The study was designed as a qualitative, explorative study using triangulation in order to support validity [24]. We used an inductive, descriptive, and explorative type of analysis, i.e., systematic text condensation (STC) developed by Kirsti Malterud based on Giorgi’s psychological phenomenological analysis [25]. The study was based on two rounds of three focus groups (i.e., six group discussions in total) with people with mental illness who use community mental healthcare services located in three different regions in Denmark.

### 2.2. 100 K for Mental Health Programme

The cycling initiative consisted of a weekly, outdoor 10 km bike ride over a period of five months. The participants were people with mental illness, and the aim for each participant was to achieve up to 100 km of cycling during the programme’s duration. The primary instructor was a member of staff who, apart from being a social–psychiatric employee, had completed special training for physical exercise for people with mental illness. This member of staff, together with the volunteers from the Danish Cyclists’ Federation, planned the route each week. The participants rode their own bikes. The bike rides took place on the same day and at the same time every week in each of the three towns where this study was carried out, with 5 to 15 participants. To participate in the outings, the participants had to be enrolled in the programme and also had to individually record how many kilometres they covered in total (data not presented). At the end of the programme, diplomas were awarded to the participants who had notched up 100 km.

### 2.3. Sampling and Recruitment

Convenience sampling was used. People with mental illness who had received standard support and care in the context of the community mental healthcare service were informed about the study orally by staff and by posters at the centre. Participants were eligible for the first baseline focus group if they expressed an interest in participating in the programme, whereas participants were eligible for the second follow-up focus group only if they had completed the programme and cycled 100 km. In practice the participants were recruited by the primary instructors, since they were familiar with the individual participants’ interests and how successfully they had completed the intervention. It was left to the primary instructors to decide whether they wanted to join in the interview. The focus groups lasted between 30 minutes to one hour. The participants all signed a written consent form and were guaranteed full anonymity.

### 2.4. Focus Groups

Altogether, six focus groups were conducted in three of the eight towns chosen from across the country, such that they covered different geographical areas. The first round of focus groups was held in May 2017 just before the “100K for Mental Health” programme kicked off, while the second round was held in October 2017 upon completion of the five-month programme.

To generate rich description and diverse views, opinions, and lived experiences from multiple perspectives, the method chosen for recording the participant experiences was focus groups. The group dynamic helped to create narratives about cycling as a group activity for people with mental illness [26]. The group discussions were conducted by the first author and built around a semi-structured interview guide, where participants were asked about their expectations, motivation, cycling experience, and the benefits of cycling, including physical, mental, and social wellbeing. Owing to the broad focus of the initiative, no questions were asked pertaining to the participants’ diagnoses or treatment.

### 2.5. Analysis

The focus group discussions were taped and transcribed verbatim. The data were analysed using systematic text condensation by the first author (H.S.) and the senior author (J.M.), a trained nurse and a trained psychologist, both female [25]. First, we separately read the transcripts several times to get a general, overall impression and with the purpose of identifying preliminary themes associated with the participants’ experience with group-based, out-door road cycling. In this step we identified preliminary themes, such as the hope of losing weight, being fitter, sleeping better, and having fun together. There were also concerns about the ability to cycle 10 km, for instance, due to inclement weather or the condition of their bikes. In the next step, H.S. and J.M. individually identified and coded meaning units representing different experiences of participants. For example, we searched for text that revealed something about the participants’ motivation for taking part and indicated what the participants had gained from the bike rides mentally, physically, and socially. The text passages identified were coded as a unit of meaning together with similar extracts that were also pertinent to their unit. Third, the content of the coded meaning units was condensed and sorted into thematic code groups (i.e., categories). In this step, the individual code was closely analysed and then we looked for further nuances within each category before linking them to the overriding themes. In the fourth and last part of the analysis, we determined the final themes and illustrated them with authentic quotations. The entire text was read through once more to re-identify themes in the text as a whole and, at the same time, ascertain to what extent the analysis answered the research questions. After the analysis was completed, an independent researcher, author S.L. (trained psychiatric nurse, female), reviewed the transcripts and performed an analysis independently. Any discrepancies between the initial analysis and the independent researcher’s analysis were discussed via multiple meetings until consensus was reached.

## 3. Results

### 3.1. Participants

The focus groups involved 25 participants spread over six discussions with three to six participants per round, including one to two members of staff. The staff participated on the request of participants, who felt too vulnerable to participate in the focus group without a member of staff present, who they knew well. The participants represented different mental health disorders, e.g., psychosis, depression, anxiety, personality disorder, and post-traumatic stress disorder. However, because the study was conducted in a non-clinical, community-based mental healthcare setting, we did not have access to participants’ backgrounds including specific diagnosis, time of diagnosis, and possible co-morbidities, for example, in one location the participants in both the first and second focus group discussions were identical, but this was not possible in the two other locations due to illness, hospitalisation, or non-completion of the 100 km (see Table 1 which lists the focus groups and numbers of participants).

### 3.2. Qualitative Findings

Four overriding themes emerged from the analysis: (1) Reinvigoration, (2) motivation through equal status, (3) group commitment without focus on illness, and (4) the value of cycling.

Table 2 provides an overview of the analysis and findings.

#### 3.2.1. Reinvigoration

In the case of the first focus group discussions, the participants’ cycling experience ranged from very little to moderate to regular cycling; however, only very few used their bikes as a means of transport. One consistent motivation factor for joining in was the desire to improve physical and mental wellbeing, including better sleep, better physical condition, and fewer negative thoughts. In the case of the second focus group discussions, many of the participants reported, unprompted, that they were sleeping better and mentioned that the bike rides made them feel tired in a natural way. The participants recounted positively their experience of physical exertion and of generally using, noticing, and discovering their bodies in a new way. One participant stated:
“*You sleep better because you’ve used your body and you’ve also got rid of lots of the waste products hanging around in your body, which are also one reason why you don’t feel very good.*”

Furthermore, according to the participants, bike rides were energising in the sense that they released energy for other purposes. Some of the participants cited this as the reason why they got out of bed or off the couch:
“*That’s another effect of sport. You can face doing all those annoying things you’re supposed to do, rather than lying around on the couch all day and smoking. You feel energised and feel like doing things again. There are two ways of feeling tired—there’s lethargy and there’s physical tired*.”

The participants emphasised the importance of getting a break from their mental issues. They especially appreciated the fact that during the outings, the conversation could revolve around anything except illness—that they had “normal conversations on an equal footing”. Some participants said that cycling essentially relieved their symptoms:
“*I also cycle to get rid of my thoughts, for example. If I have thoughts that are bugging me, they go away when I get on my bike.*”

For some participants, cycling improved their physical fitness and made them feel healthier. The participants pointed out that they enjoyed joining in and that they could notice an improvement in their body and muscle strength. Some participants noticed an improvement in their breathing and that uphill cycling—which seemed an insurmountable challenge at first—gradually felt more manageable.

#### 3.2.2. Motivation through Equal Status

As far as the participants’ motivation to cycle was concerned, the role of the staff and the volunteers was seen as essential. The participants emphasised how helpful it was to receive a text message reminding them that “we’re going cycling today”, and how the staff were particularly good at demonstrating more than once how to cycle, giving them a push, exerting mild pressure on individuals, and giving them a “gentle shove”:
“*I think it’s because she’s easy to talk to and she gives us information. And her friendly way of coming up to you, patting you on the shoulder and asking, ‘Did you enjoy that?*’” 

The participants also pointed out their particular appreciation of the active participation by staff and the volunteers, who came across as role models and as having an equal status, with statements such as:
“*The staff also do it (participate in the cycling) and come across as more human. I like the fact that the staff also join in the exercise*” and“*You couldn’t tell who was a social worker and who was an ordinary person.*”

The energy and enthusiasm of the staff and the volunteers, coupled with their faith in the participants’ ability to complete the bike rides, were cited as the main reasons why the participants got on their bikes and felt a sense of personal responsibility:
“*It’s because there’s someone like him with us that I feel like doing more, like helping with the arrangements so he doesn’t get stuck with everything. One of the things for me is pulling together as a team.*”

#### 3.2.3. Group Commitment without Focus on Illness

Being part of a group commitment was the first thing that sprang to mind when the participants were asked about the advantages of taking part in the regular bike rides. Being part of a group activity meant a commitment, in that the participants knew that someone was waiting for them. That got them out the door more often, even on days when they felt sluggish and the weather was bad.
“*If you’ve arranged to do something with someone else, it’s easier to get yourself off the couch than if you have to motivate yourself to do it and it’s pouring with rain You know there’s someone waiting for you just round the corner and you just have to go out and do it.*”

All the group discussions included accounts of outings in rainy, windy weather or outings which turned out to be too long and ended with sore legs and bottoms. These stories were related humorously and had clearly been told many times over. Humour and the feeling of common achievement and purpose helped motivate the participants to meet up. Across the three group discussions the participants described cycling as an activity that united the group and that revolved around something other than being ill—an activity which helped the participants to counteract social isolation and solitude. One participant commented:
“*The social side is also a big part of it for me. So often I’ve felt down and lonely after a weekend. With the bike rides you know someone’s waiting for you. I’ve joined in the rides several times. They really buck up your mood for the rest of the day.*”

#### 3.2.4. The Value of Cycling

The participants did not see cycling as the only sports activity that energised and conveyed a sense of community and solidarity. However, they did describe cycling as having special attributes. One of the benefits pointed out by the participants was the fact that it was possible to chat while biking without having to make eye contact, providing a more informal, less strenuous setting. Another particular feature of cycling highlighted by the participants was the fact that there was no need to bear one’s own body weight, unlike with walking or running. As one social worker from a centre explained:
“*We have a running group, but of course there are some people who can’t join in or maybe never will be able to. You have to be fit to a certain extent, unlike cycling, which isn’t the same and is easier. If you’re sitting on a bike, you can join in to varying degrees and you are close enough to talk with the others, but you can also just be part of it without saying anything and without it coming across as strange. Group cycling is like that.*”

Finally, but just as importantly, according to the participants, cycling got them outdoors and stimulated their senses. The participants mentioned their appreciation of the green leaves on the trees, the scents in the air, and the wind in their faces making their cheeks turn red and making them notice the difference between cycling and sitting in front of the TV or computer. One of the participants talked about having to get accustomed to being outdoors:
“*Being in fresh air rather than a smoky atmosphere is something you have to get used to, so it’s about habits and changing them. It was really good to get out and breathe in so much fresh air rather than smoke-filled air from cigarettes.*”

## 4. Discussion

In this study which seeks to describe experiences with group-based outdoor cycling for people with mental illness, we found that outdoor cycling, from the perspective of the participants, is associated with a unique feeling of personal mastery, equal status, solidarity, community, and healing sensory experiences, including the sensation of fresh air, wind, and rain.

A main finding in the study was the supportive function of the staff, who provided encouragement, cheered up the participants, and sometimes gave them a fond nudge to get them going. Their support was crucial for giving the participants a sense of increased personal mastery and faith in being able to complete the outings. This finding is especially interesting, as low self-efficacy—here, a lack of belief in one’s ability to engage in physical activity—is one of the obstacles to participation in physical exercise [9,14,15,16]. Overall, the findings in this study indicate that the participants gained greater self-efficacy, backed by verbal persuasion in the form of the staff’s support and encouragement [27], together with vicarious experience, where the participants become role models for each other. They became motivated through observing how others with the same difficulties as themselves joined in the regular bike rides.

Apart from improved self-efficacy, this study also demonstrates the central importance of participation by the members of staff, who cycled alongside the patients; this helped to increase the patients’ motivation and ensured that they kept up the activity [28]. In this study, the participants felt there was a closer bond between the patients and staff when they cycled together. The collaborative activity engendered informal contact, with the contact taking on a more relaxed nature, since it involved a common activity. Not only did it serve to make the contact informal, but also placed relationships between the staff and the people with mental illness on an equal footing, thus contributing to the personal recovery process, where contact with others and a feeling of belonging was one of several important areas in the recovery process [29]. Establishing a trustful patient–professional relationship is fundamental in psychiatry, as the alliance between the patient and the professional is the very essence of the therapeutic process [30,31].

It is interesting to note that the participants in the current study experienced a collaborative relationship with peers and professionals (as they worked together to achieve a mutually agreed upon goal) in a context not centred on medicine. In contrast, it has recently been shown that the feeling of being assessed from a distance by health professionals constitutes a barrier to recovery [32]. Whether and how normal physical activities and participating in sports outside the hospital setting may constitute a unique potential for recovery in the mental healthcare setting warrants further studies. Moreover, a noteworthy finding in the current study was the participants’ experience that the bike rides improved their sleep and made them feel tired in a natural way. This is in accordance with a recent meta-analysis [33] showing the beneficial effect of exercise on sleep quality in individuals with mental illness. As such, further research determining the efficacy of exercise, specifically outdoor activities, seems warranted and may support the health-promoting role of exercise in people living with mental illness.

A surprising finding in the current study was that the participants experienced a sense of relief and integrity from exercising and engaging with other people side by side (on a bike), instead of face to face with eye contact. As such, the current study challenges existing research promoting the therapeutic effects of frequent and/or prolonged eye contact [34] as a key factor in motivating clients to engage in the therapeutic process, including potentially leading to more positive appraisal of the self and others. In the current study, participants did not experience the lack of prolonged eye contact as a decrease in the natural tendency to need reassurance, but instead experienced that a burden was lifted, including the pressure to be polite and to smile. More work is needed to understand the potential therapeutic impact of not having to make eye contact in exercise-based recovery/rehabilitation.

One aspect of cycling highlighted by the participants in this study was not having to bear one’s own body weight, which is necessary in other forms of exercise. A walk, for example, can be problematic for people who are overweight, as excess body weight leads to discomfort, such as shortness of breath and pain in the musculoskeletal system, which is not under the same strain when on a bicycle [9]. Another particular feature is that the participants moved around outdoors and used their senses, which lent the experience yet another facet. Participants from a psychiatric centre involved in the “Walking Back to Health” project felt a sense of nature as something very special. Apart from finding it refreshing to be outdoors, they learned a great deal, including a better understanding of flora and fauna [35]. Moreover, Kondo et al. [36] recently provided evidence that spending time in outdoor environments, particularly those with green spaces, may reduce the experience of stress, and ultimately improve health (in the general population). James et al. [37] also recently described how outdoor activities, specifically an adapted hiking programme, offered unique social opportunities for people with disabilities and volunteers.

Our study has a variety of strengths and limitations that should be taken into consideration. The use of qualitative methods provided a broad understanding but limited specific inferences, and the people interviewed for this study may not reflect the perspectives of all people with mental illness. One of the strengths of the current study is the level of credibility due to the number of collaborative sessions the authors did to test ideas and the interpretation [38]. A weakness is that the method used and the moderate number of overlapping participants from the first to the second focus group discussions did not allow the authors to analyse changes as such. However, the use of convenience sampling yielded a diverse and thus more representative sample, which was fortunate in terms of validity in that more nuances were added. The size of the focus groups in four out of six cases was lower than the recommended size of six to eight [39]. However, even the group discussions including only three participants were characterized by a dynamic exchange of experiences and attitudes. Another strength is that the three towns and cities were geographically far enough apart to take into account any local discrepancies. The themes that were identified occurred in all six focus group discussions, providing internal and external validity and indicating the transferability of the findings. A limitation of the study, which impacts transferability, is that we did not include those participants who did not complete the 100K. It may therefore be assumed that the participants in this study were particularly motivated and enthusiastic. Consequently, the findings of the study alone can be generalised to a corresponding group of motivated individuals [38]. Moreover, although it was stated repeatedly that participation in the study was completely voluntary and that refusal or withdrawal from participation would not impact on care, it cannot be ruled out, that some service users may have found it difficult to refuse to participate in or withdraw from the study because of reliance on the amity of professionals.

Further research on what it means for people with mental illness and their inhibitions about participation in sports to be together in an exercise or physical activity group is warranted. It would be particularly interesting to examine the implications for the individual participant’s experience of recovery, including how physical exercise may be used as a means to improve recovery in association with and in psychiatric wards and community mental health services. This study indicates that community-based outdoor cycling has substantial therapeutic potential for people with mental illness, who are otherwise unwilling or unable to participate in centre-based, indoor activities. In this regard, a recently published protocol for a systematic review [40] indicates that more knowledge will be gained in the future in relation to our understanding of the needs and expectations of patients with serious mental illness in relation to community-based group physical activity. This is expected to support the design of more acceptable and effective future programmes in the context of health promotion in mental health care.

## 5. Conclusions

The participants in this study associated outdoor cycling with improved physical, mental, and social wellbeing related to a feeling of personal mastery, equal status, solidarity, community, and healing sensory experiences, including the sensation of fresh air, wind, and rain. The support and companionship of the staff members were pivotal for the participants’ experience and motivation to persist in the programme. The impact of exercise on the relationship between patients and staff merits further research and interventional studies. Similarly, it is necessary, using randomised clinical trials, to answer the question of whether outdoor cycling can help to reduce the mental health burden and increase exercise self-efficacy in people with mental illness.

## Figures and Tables

**Table 1 ijerph-16-00528-t001:** Focus group discussions and participants.

Participants	Total	First Focus Groups (May 2017)			Second Focus Groups (Oct. 2017)		
		Town 1	Town 2	Town 3	Town 1	Town 2	Town 3
Number (*n*)	25	3	3	6	4	6	3
Male/Female	9/16	0/3	1/2	1/5	0/4	5/1	2/1
Participant/Staff	20/5	2/1	3/0	5/1	3/1	6/0	1/2
Repeat participants *	4				3	0	1

* People who took part in both the first and second focus group discussion.

**Table 2 ijerph-16-00528-t002:** Overview of the analysis and findings.

Units of Meaning (Sample)	Categories	Theme
Doing something good for your body Feeling fitter Sleeping better Feeling physically tired rather than lethargic Better breathing Cycling suppresses annoying thoughts Cycling relieves the symptoms Cycling energises Cycling highlights the human qualities of the staff members, who demonstrate commitment Being in a group in a context not related to illness Normal conversation on an equal footing A fond nudge in the right direction A well-meaning shove Gentle pressure Believing in your ability Being bolder Visible and much appreciated effort by the staff Will to make an extra effort Escaping loneliness Getting out of bed Being together with like-minded people It’s okay to be a bit weird Friendly teasing Everyone can ride a bike, but not everyone can run, play football, etc. Don’t have to bear your own weight No need for eye contact Serious but also casual conversation while biking Being outdoors Experience of smells and colours	Improvement of physical condition	Reinvigoration
	Improvement of mental condition	
	New feeling of tiredness	
	Staff as role models	Motivation through equal status
	Staff as motivators	
	Pulling together as a team	
	Others are waiting	Group commitment without focus on illness
	Safe haven—no bullying	
	Humour	
	No strain if you are overweight	The value of cycling
	Lack of eye contact	
	Nature experience	

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
