# Peer review of "Experiences with Participation in a Supervised Group-Based Outdoor Cycling Programme for People with Mental Illness: A Focus Group Study"

_ijerph, 2019, doi:10.3390/ijerph16040528_

Round 1
Reviewer 1 Report
The matter of the manuscript is interesting. However, the manuscript has some determinant weakness described in the attached document.

Author Response
Please see uploaded Word document

Reviewer 2 Report
This manuscript aimed to study whether the experiences of patients with mental illness who took part in group-based outdoor cycling, including their experience of physical, social and mental wellbeing. Although is an interesting study, there are several problems with this manuscript, which I discuss below:
1) Reduced sample size (n=25), which entails scarce statistical power. Moreover, authors did not control important confounding variables such as socio-demographic variables (e.g age, marital status, educational level…) or other characteristics of the mental disorder (e.g. type, drug use, age-of-onset, current status of mental disorder…).
2) Absence of an appropiate control group to compare the success of their intervention (e.g. motivated patients vs participants with low motivation to participate in the study).
3) Absence of objective health measures (blood analytics, blood preassure…).
4) Please state the inclusion/exclusion criteria explicitly for this study.
5) The association between mental disorders (all???) and weight problems. In fact, this association is innaccurate. Hence, it would necessary to template this kind of sentence (e.g. It is well documented that people with severe mental illness have a shorter life expectancy compared to the general population [1, 2]) based on scarce empirical studies.
Finally, I would recommend to conduct an objective study in order to demonstrate their main objective.
Author Response
Please see uploaded Word document

Reviewer 3 Report
The work entitled “Experiences with Participation in a Supervised, Group-based, Outdoor Cycling Programme for People with Mental Illness: A Focus Group Study " is of great interest in the stress field. The research is very stimulating; it contains new scientific knowledge and provides comprehensive information for further development of this productive line of research. The data are presented in a clear and easy-to-understand fashion, and the paper is, principally, well-argued and clearly worthy of publication. However, I have some comments to make that should be addressed before I recommend this manuscript for publication:
The introduction section is well written and provides insightful previous literature about the research, nonetheless I have some concerns. The authors focus on the physical consequences derived of mental health problems but there are not a lot of information about the impact of mental health in subjective well being or quality of life which is actually the main goal of the present research. Thus, did the authors collect some information about the physical impact of the programme. Also, considering that the goal of the study is analyzing the perceived impact of the programme including subjective perception of well being and quality of life, the following recent studies could add value in order to enhance the introduction and later discuss the results (Fonseca-Pedrero, Inchausti, Pérez-Gutiérrez, Aritio Solana, Ortuño-Sierra and Sánchez-García, 2018; Gardsjord, Romm, Røssberg, Friis, Barder, Evensen et al., 2018; Ortuño-Sierra, Aritio-Solana, Chocarro de Luis, & Fonseca-Pedrero, 2017; Rowe, Masters, You, Burrows, Lai, Lautenschlager, 2018; Tafoya, Aldrete-Cortez, 2018).
First, concerning the participants section, I think that it would be advisable to deeper explain the sample characteristics. For instance, did the authors collect data about the previous time of psychotherapy or about those who suffered a relapse? If not this should be comment in the limitations section. Also, I think that information about participants’ gender and age should be included in the participants section.
Related to the previous comment, did the author consider the possibility of analyzing the data attending to gender and age? I understand that this a qualitative study but this could be an added value. Also, did the authors check for qualitative data about for instance, quality of life or well-being, this would have been an extra value for the study. If not these could be introduced in the limitations and prospective sections.
Also, more information about the training could be added. For instance, how many days a week did the participants train? Was the training perform in group or only with the instructor?
Another question that remains some how unclear is the cathegorization system. Did the authors check for interjudge reliability in order to assure the reliability of the observations?
Author Response
Please see uploaded Word document

Reviewer 4 Report
Dear authors, this study protocol constitutes a valuable contribution to our knowledge about social cycling and mental health. As it stands, the underpinning theory is not clear to me. A number of social support theories have been developed and I suggest embedding this into your manuscript. I added more detailed comments into the PDF file and suggest reviewing current literature of exercise and mental health. References have been provided in the comments in PDF file.
I suggest publication pending minor revisions. Comments have been added as sticky notes.
On the positive side, this article is well structured and findings will add to the new knowledge using social cycling as an option to support and treat mentally ill people.

Author Response
Please see uploaded Word document

Round 2
Reviewer 1 Report
I appreciate the efforts of the authors and there has been an improvement of the manuscript but the main points have not been resolved.
Author Response
Round 2 - Response to Reviewer 1 Comments
We thank Reviewer 1 for the review comments of the revised version of our paper entitled: Experiences with Participation in a Supervised, Group-based, Outdoor Cycling Programme for People with Mental Illness: A Focus Group Study.
Below, please see our response to the Reviewer 1’s comments.
Reviewer 1 writes: I appreciate the efforts of the authors and there has been an improvement of the manuscript, but the main points have not been resolved. Reviewer 1 furthermore notes that we must improve the manuscript in relation to these two points: (1) research design, and (2) description of methods.
Response: Because Reviewer 1 doesn’t provide any specific/detailed comments, we can’t know exactly which changes Reviewer 1 is suggesting and for what reason, and therefore we find it very difficult to respond. Moreover, none of the other three reviewers suggest that we need to address the design of the study and the methods further. Therefore, we stick to our previous response to reviewer 1 in Round 1, namely that we disagree that there is anything unfortunate about our choice of design. Instead, we believe that that the use of qualitative research methodology in this study was essential to fulfill the study’s overall aim, i.e. explore meanings of a social phenomenon as experienced by individuals themselves, in their natural context.
Some additional methodological reflections: The credibility (validity) in qualitative research deals with how congruent the findings are with reality. In the current study, the use of convenience sampling yielded a diverse and thus relatively representative sample. Moreover, data were collected in three different locations and the identified themes emerged across all six focus group discussions and in all study sites. Furthermore, and to ensure reflexivity and confirmability (objectivity) we used an independent researcher, who reviewed the transcripts and performed an analysis independently. This was to ensure any discrepancies between the initial analysis and the independent researcher’s analysis, and the findings presented were discussed via multiple meetings until consensus was reached. We acknowledge that it is very important to describe the methods in detail (as a prototype model), and we believe that we have already done so in section 2.5. (see p 3,4). We also believe that we admit to the limitations of our study and have described these in the Discussion section where we also recommend further research.
H owever, to comply with Reviewer 1 (i.e. do our best to meet his request), and to ‘admit’ to this being a qualitative study, we have inserted the following sentence in the Discussion (see p 8): “The use of qualitative methods provided a broad understanding but limits specific inferences, and the people interviewed for this study may not reflect the perspectives of all people with mental illness.”
Reviewer 2 Report
Although authors have tried to convince me about a qualitative research, there are several problems with this manuscript:
1) Reduced sample size (n=25), which entails scarce statistical power. Moreover, authors did not provide important confounding variables such as socio-demographic variables (e.g age, marital status, educational level…) or other characteristics of the mental disorder (e.g. type, drug use, age-of-onset, current status of mental disorder…).
2) Absence of an appropiate control group to compare the success of their intervention.
3) Absence of objective health measures (blood analytics, blood preassure…).
Author Response
Round 2 - Response to Reviewer 2 Comments
We thank Reviewer 2 for the review comments of the revised version of our paper entitled: Experiences with Participation in a Supervised, Group-based, Outdoor Cycling Programme for People with Mental Illness: A Focus Group Study.
Below, please see our response to the Reviewer 2’s comments.
Point 1: 1) Reduced sample size (n=25), which entails scarce statistical power. Moreover, authors did not provide important confounding variables such as socio-demographic variables (e.g age, marital status, educational level…) or other characteristics of the mental disorder (e.g. type, drug use, age-of-onset, current status of mental disorder…).
Point 2: Absence of an appropiate control group to compare the success of their intervention.
Point 3: Absence of objective health measures (blood analytics, blood preassure…).
Response:
As this was a qualitative study we have not been able to comply with Reviewer 2’s requests – nor do we regard them as relevant. We continue to believe that the use of qualitative methodology was and is the (only) appropriate choice when the aim is to investigate the meaning of social phenomena as experienced by the people themselves it was the case in the current study. As such, we fundamentally disagree with Reviewer 2 that a larger sample, a control group and objective outcomes would have improved the study. Qualitative research provides “evidence beyond measures and numbers” as stated by Malterud in The Lancet in 2001 (Lancet 2001; 358: 397–400). Reviewer 2’s comments reflect a belief that knowledge can only be defined as facts that can be controlled, measured, counted, and analyzed by statistical methods. This, however, according to Malterud (2001) represents “a confined access to clinical knowledge”.
Reviewer 3 Report
No further comments.
Author Response
Thank you for no further comments in round 2 review
Reviewer 4 Report
The authors have addressed the majority of comments. I would like to point towards embedding a research methodology to the manuscript and have provided suggestions in the uploaded file.

Author Response
Round 2 - Response to Reviewer 4 Comments
We thank Reviewer 4 for the review comments of the revised version of our paper entitled: Experiences with Participation in a Supervised, Group-based, Outdoor Cycling Programme for People with Mental Illness: A Focus Group Study.
Below, please see our response to the Reviewer 4’s comments.
Point 1: P. 2 - Review and not seveiw
Response 1: Thank you, this has been corrected – se manuscript p. 2, line 60
Point 3: P.3. reads much better now and P.4., ok, much better
Response 3: Thank you
Point 4, 5 and 6: no need to include the initials instead - the authors...
Response 4, 5 and 6: We appreciate Reviewer 4’s suggestion. However, because we believe that it is of importance to clarify the specific roles of the involved researchers in relation to the analysis, we would like to stick to the use of initials.
Point 2: What methodology underpins your research? Methods are different to methodologies - think of Ethnography / Narrative / Penomenological / Grounded Theory...
Refer to source: Creswell, J W 2013, Qualitative Inquiry and Research Design: Choosing Among Five Approaches, 3rd edn., SAGE publications
Response 2: Thank you for the suggestion to refer to Creswell’s book. We used an inductive, descriptive and explorative type of analysis, i.e. systematic text condensation (STC) STC is developed by Kirsti Malterud and is based on Giorgi’s psychological phenomenological analysis. This information has been added to the Methods section, (see p 2,3): “We used inductive, descriptive and explorative type of analysis, i.e. systematic text condensation (STC) developed by Kirsti Malterud based on Giorgi’s psychological phenomenological analysis.”